# A recurrent kinase domain mutation in *PRKCA* defines chordoid glioma of the third ventricle

Benjamin Goode[1], Gourish Mondal[1], Michael Hyun[1], Diego Garrido Ruiz [2], Yu-Hsiu Lin[3], Jessica Van Ziffle[1,4], Nancy M. Joseph[1,4], Courtney Onodera[4], Eric Talevich[4], James P. Grenert[1,4], Iman H. Hewedi[5], Matija Snuderl[6], Daniel J. Brat[7], Bette K. Kleinschmidt-DeMasters[8], Fausto J. Rodriguez [9], David N. Louis[10], William H. Yong[11], M. Beatriz Lopes[12], Marc K. Rosenblum[13], Nicholas Butowski[14], Tarik Tihan[1], Andrew W. Bollen[1], Joanna J. Phillips[1,14], Arun P. Wiita [2,3], Iwei Yeh[1,4], Matthew P. Jacobson[2], Boris C. Bastian[1,4], Arie Perry [1,14] & David A. Solomon [1,4]

Chordoid glioma is a rare brain tumor thought to arise from specialized glial cells of the lamina terminalis along the anterior wall of the third ventricle. Despite being histologically low-grade, chordoid gliomas are often associated with poor outcome, as their stereotypic location in the third ventricle makes resection challenging and efficacious adjuvant therapies have not been developed. Here we performed genomic profiling on 13 chordoid gliomas and identified a recurrent D463H missense mutation in *PRKCA* in all tumors, which localizes in the kinase domain of the encoded protein kinase C alpha (PKCα). Expression of mutant *PRKCA* in immortalized human astrocytes led to increased phospho-ERK and anchorage-independent growth that could be blocked by MEK inhibition. These studies define *PRKCA* as a recurrently mutated oncogene in human cancer and identify a potential therapeutic vulnerability in this uncommon brain tumor.

[1] Department of Pathology, University of California, San Francisco, CA 94143, USA. [2] Department of Pharmaceutical Chemistry, University of California, San Francisco, CA 94158, USA. [3] Department of Laboratory Medicine, University of California, San Francisco, CA 94107, USA. [4] Clinical Cancer Genomics Laboratory, University of California, San Francisco, CA 94115, USA. [5] Department of Pathology, Faculty of Medicine, Ain Shams University, Cairo 11591, Egypt. [6] Departments of Pathology and Neurology, NYU Langone Medical Center, New York, NY 10016, USA. [7] Department of Pathology, Northwestern University Feinberg School of Medicine, Chicago, IL 60611, USA. [8] Department of Pathology, University of Colorado at Denver, Aurora, CO 80045, USA. [9] Department of Pathology, Johns Hopkins University School of Medicine, Baltimore, MD 21287, USA. [10] Department of Pathology, Massachusetts General Hospital, Harvard Medical School, Boston, MA 02114, USA. [11] Department of Pathology, David Geffen School of Medicine, University of California, Los Angeles, CA 90095, USA. [12] Department of Pathology, University of Virginia School of Medicine, Charlottesville, VA 22908, USA. [13] Department of Pathology, Memorial Sloan Kettering Cancer Center, New York, NY 10065, USA. [14] Department of Neurological Surgery, University of California, San Francisco, CA 94143, USA. Benjamin Goode and Gourish Mondal contributed equally to this work. Correspondence and requests for materials should be addressed to D.A.S. (email: david.solomon@ucsf.edu)

Chordoid glioma of the third ventricle is characterized by its stereotypic location in the anterior third ventricle and its chordoid cellular architecture composed of glial fibrillary acidic protein (GFAP)-positive tumor cells embedded within a myxoid matrix, often accompanied by a dense lymphoplasmacytic inflammatory infiltrate[1–3]. Due to their intraventricular location, they often obstruct flow of cerebrospinal fluid, leading to symptoms associated with obstructive hydrocephalus, such as intracranial hypertension, headache, nausea, vomiting, ataxia, and cognitive changes. Despite being histologically low-grade (classified as grade II in the 2016 WHO Classification of Tumors of the Central Nervous System) and well circumscribed without substantial invasion of surrounding brain tissue, chordoid gliomas are associated with substantial morbidity and mortality. Their proximity and adherence to critical regional structures makes them difficult to safely resect. While gross total resection remains a goal and can be curative, it can result in hypothalamic dysfunction or peri-operative complications including death[4, 5]. Subtotal resection is often accompanied by tumor recurrence. Optimal adjuvant therapy for subtotally resected or recurrent tumors has not been established.

Chordoid gliomas were recently demonstrated to share expression of the homeobox transcription factor TTF-1 (also known as NKX2.1) with the organum vasculosum of the lamina terminalis, suggesting a cellular origin of this neoplasm from these specialized ependymal cells located along the anterior wall of the third ventricle[6]. However, the molecular alterations that drive these tumors are unknown[7]. Here we report genomic analysis on a cohort of chordoid gliomas that was performed with the goal of identifying new diagnostic and prognostic biomarkers, as well as potential targets for molecularly tailored therapy for these rare brain tumors.

## Results

**Chordoid glioma patient cohort.** To investigate the molecular pathogenesis of this rare brain tumor entity, we assembled a cohort of archival tissue specimens from 13 patients diagnosed with chordoid glioma of the third ventricle at medical centers around the world. Clinical features of these 13 patients are listed in Supplementary Table 1. The 6 male and 7 female patients ranged in age at time of biopsy or resection from 34–67 years (median 48 years). All tumors were located in the anterior third ventricle and ranged in size from 2 to 6 cm. Representative imaging features are shown in Fig. 1a and Supplementary Figure 1. Pathologic review confirmed the diagnosis of chordoid glioma for each case. Representative histologic and immunohistochemical features are shown in Fig. 1b and Supplementary Figures 2 and 3. Patient outcomes ranged from early postoperative death from surgical complications to mortality from tumor recurrence within one year following subtotal resection to long term recurrence-free survival.

**PRKCA D463H mutation defines chordoid glioma.** We performed targeted next-generation sequencing as previously described on genomic DNA isolated from the 13 tumors, as well as matched normal tissue when available (two cases)[8]. In each tumor, a G>C transversion mutation in the PRKCA gene was identified causing a c.1387G>C, p.D463H substitution (reference transcript NM_002737; Supplementary Table 2). This missense mutation was verified to be somatic in both tumors with matched normal tissue available for sequencing (CG-UCSF-1 and CG-UCLA-1; Supplementary Figure 4). The mutant allele frequency for this PRKCA D463H variant ranged from 12 to 42% in the 13 tumors, consistent with a somatic heterozygous mutation in all cases. The cases with the highest PRKCA mutant allele frequencies had genomic DNA that was isolated from areas histologically visualized to contain a high tumor cell content. The cases with lower PRKCA mutant allele frequencies had genomic DNA that was isolated from areas that were visualized to contain a more abundant lymphoplasmacytic inflammatory cell infiltrate. The PRKCA locus is located at chromosome 17q24.2. No chromosomal gains, losses, or copy-neutral loss of heterozygosity involving this chromosomal locus were identified in any of the tumors (Supplementary Figure 5, Supplementary Table 3). Therefore, these data suggest that this PRKCA D463H mutation is likely to be a clonal alteration in chordoid gliomas (i.e., present in all tumor cells), indicating that it is probably an early or initiating event in tumorigenesis.

Besides PRKCA mutation, no somatic nonsynonymous mutations, amplifications, deletions, or rearrangements were identified in any of the other 478 genes sequenced in the 13 chordoid gliomas (Supplementary Data 1). Ten of the tumors demonstrated a balanced diploid genome, while the remaining three tumors contained isolated whole chromosome or arm-level gains and losses, without focal regions of amplification or deletion (Supplementary Figure 5, Supplementary Table 3). Specifically, there were no alterations identified involving NF2 or RELA, indicating that chordoid gliomas are genetically distinct from most spinal and supratentorial ependymomas[9]. Additionally, there were no alterations identified involving IDH1, IDH2, TP53, ATRX, TERT, CIC, FUBP1, and NOTCH1, indicating that chordoid gliomas are genetically distinct from the vast majority of diffuse lower-grade gliomas in adults[10]. Also, there were no alterations identified involving TSC1 or TSC2, indicating that chordoid gliomas are genetically distinct from subependymal giant cell astrocytomas[11], another subtype of intraventricular glioma that may show morphologic overlap with chordoid glioma of the third ventricle (Supplementary Figure 2e).

PRKCA encodes protein kinase C alpha (PKCα), a cytoplasmic serine/threonine kinase whose activity is modulated through calcium and diacylglycerol binding domains (Fig. 1c). Only 95 tumors with confirmed somatic nonsynonymous mutations of PRKCA are present amongst the 30,367 tumors with available sequencing data for PRKCA in the version 81 release of the Catalogue of Somatic Mutations in Cancer (COSMIC) database (Supplementary Data 2). The mutations in these 95 tumors are scattered throughout the gene without clustering in the kinase or other functional domain, and none are affecting codon D463 as seen in chordoid gliomas (Supplementary Figure 6). These 95 tumors are predominantly from cancer types with conspicuously high somatic mutational burden such as melanoma and microsatellite unstable gastrointestinal carcinoma, suggesting that a majority of these PRKCA variants in tumors other than chordoid gliomas are likely to be passenger or bystander mutations.

**Structural modeling of mutant PKCα.** Amino acid D463 within the kinase domain of PKCα functions as the proton acceptor during the ATP hydrolysis reaction and is highly conserved from yeast to humans (Fig. 1d). Structural modeling of PKCα illustrates that substitution of aspartate with histidine at this codon (D463H) results in a roughly isosteric side chain that contains a proton acceptor in approximately the same location of one of the oxygens in aspartate, if the imidazole ring is not doubly protonated (Fig. 1e and Supplementary Figure 7). However, the proton affinities of aspartate and histidine are markedly different (pKa of approximately 4.0 and 6.5, respectively), and the histidine side chain is commonly, but not always, doubly protonated at physiological pH. This substitution is therefore likely to alter the activity of the kinase function by perturbing the ATP hydrolysis

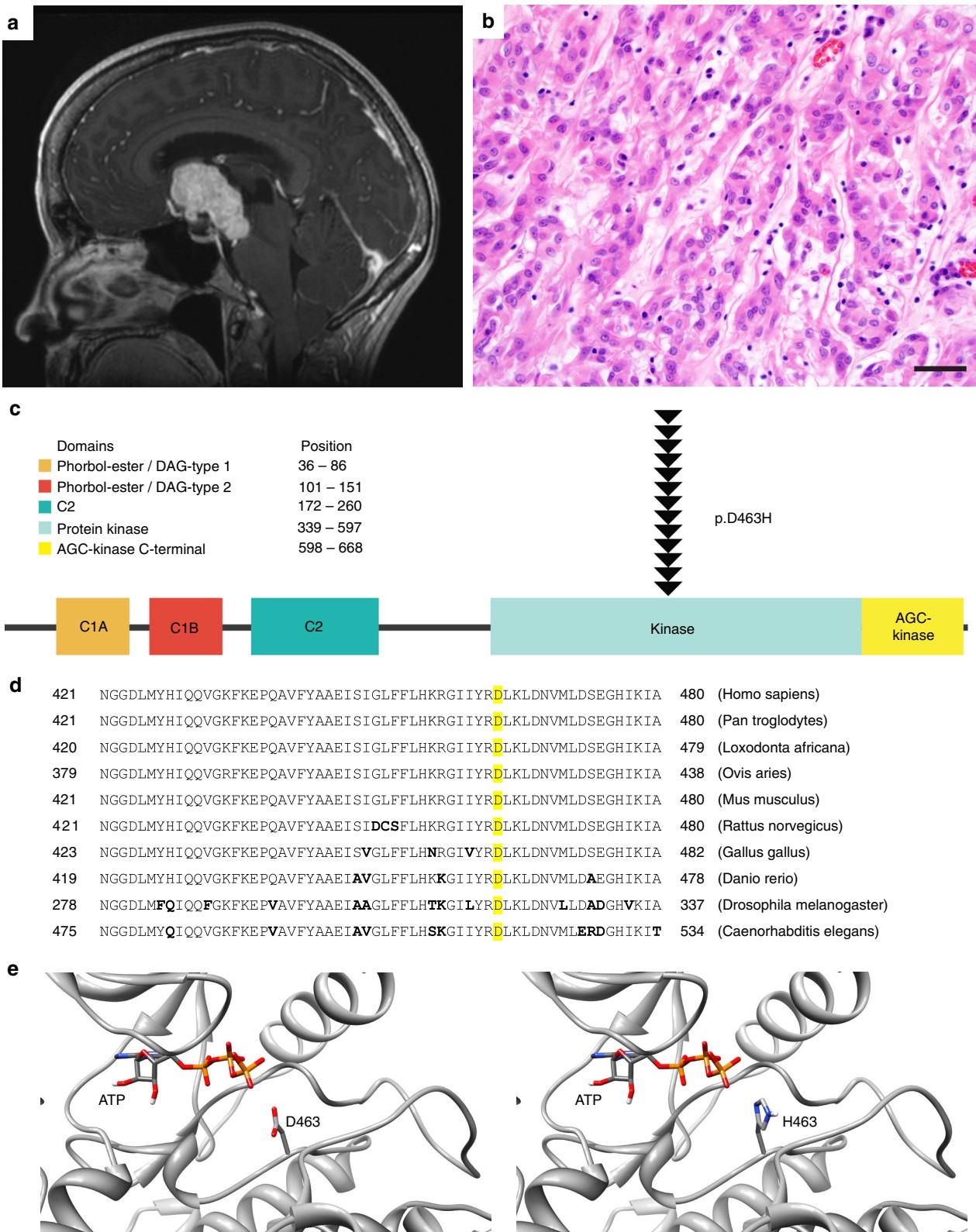

reaction. Prior in vitro biochemical analysis of PKCα has shown that synthetic isoforms harboring alanine or asparagine substitution at codon 463 (D463A or D463N) are kinase dead and lack the ability to phosphorylate a peptide substrate relative to the wildtype protein[12]. However, the functional consequence of the aspartate to histidine substitution at this codon found in chordoid gliomas (D463H) has not been evaluated to date, and we are unaware of any other protein kinases or pseudo-kinases with histidine at this position in the enzymatic pocket.

**PRCKA D463H is an oncogenic mutation that drives anchorage independent growth.** To determine the functional effects of this *PRKCA* mutation during tumorigenesis, we generated a lentiviral expression vector for the human *PRKCA* cDNA. We then used site-directed mutagenesis to introduce the D463H mutation observed in chordoid gliomas, and additionally generated a separate expression vector harboring a D463A mutation. As the amino acid side chain of alanine cannot function as a proton acceptor, we hypothesized that this *PRKCA* D463A construct may function as a kinase-dead mutant. Immortalized human astrocytes were transduced with lentivirus and then grown in soft agar to assay anchorage-independent growth, a classic measure of cellular transformation[13]. Expression of wildtype *PRKCA* led to only a small number of colonies, whereas D463H mutant *PRKCA* yielded numerous colonies, indicating a potent oncogenic effect of this mutation (Fig. 2a). No colonies were observed in immortalized human astrocytes after infection with empty vector or D463A mutant *PRKCA*. Lentiviral expression of D463H mutant *PRKCA* was also sufficient to drive anchorage-independent growth of NIH-3T3 cells (Supplementary Figure 8). These results indicate that D463H is an oncogenic, gain-of-function mutation.

**PRKCA D463H mutation induces activation of the MAP kinase signaling pathway.** To study the mechanisms by which *PRKCA* D463H mutation drives tumorigenesis of chordoid gliomas, we transiently overexpressed wildtype and mutant *PRKCA* in 293 T cells and immortalized human astrocytes. Reverse transcription-PCR analysis after ectopic expression confirmed expression of wild-type and mutant *PRKCA* transcripts (Supplementary Figure 9). No differences in phosphorylated ERK were observed in either cell type during in vitro growth in media supplemented with 10% fetal bovine serum (Fig. 2b). In addition, no increase in phosphorylated MARCKS, a known substrate of PKCα kinase activity, was observed (Fig. 2b). However, when stably transduced immortalized human astrocytes were serum starved for 24 h, a significant increase in phospho-ERK was observed for the D463H mutant relative to the wildtype and D463A mutant isoforms (Fig. 2c). This was accompanied by a less robust increase in phospho-Akt after serum starvation (Fig. 2c), presumably related to the increased proliferative signaling through the MAP kinase pathway. As opposed to the overexpression of PKCα seen after transient transfection (Fig. 2b), a significant increase in PKCα levels was not observed after stable

lentiviral transduction in immortalized human astrocytes (Fig. 2c), possibly due to only low levels of overexpression or auto-regulatory feedback mechanisms controlling the expression of PKCα.

Next we performed immunohistochemistry for phospho-ERK on the cohort of chordoid gliomas with *PRKCA* D463H mutation. In each of the six chordoid gliomas that were assessed by immunohistochemistry, robust staining for phospho-ERK was observed, which was specific to the tumor cells and not present in non-neoplastic endothelial cells and admixed inflammatory cells (Fig. 3). The high level of phospho-ERK expression in chordoid gliomas was equivalent to that seen in other glioma types harboring known genetic alterations within the MAP kinase signaling pathway, including pleomorphic xanthoastrocytoma with *BRAF* V600E mutation and pilocytic astrocytoma with *KIAA1549-BRAF* gene fusion (Fig. 3). Minimal phospho-ERK staining was observed in sections of normal brain as well as other glioma types harboring genetic alterations not associated with MAP kinase pathway activation (Fig. 3), including clear cell ependymoma with *RELA* gene fusion, which activates the nuclear factor-kappa B signaling pathway. Thus, *PRKCA* D463H mutation is likely to drive chordoid glioma, at least in part, by activation of the MAP kinase signaling pathway, although this may be a downstream consequence rather than direct phosphorylation by mutant PKCα.

**MEK inhibition blocks the oncogenic effects of mutant PRCKA.** We next tested the hypothesis that MEK inhibition might be an effective therapy for chordoid glioma of the third ventricle. Immortalized human astrocytes transduced with D463H mutant *PRKCA* alongside a series of malignant glioma cell lines harboring wildtype *PRKCA* alleles were grown in soft agar containing either DMSO vehicle or 5 μM trametinib, a small molecule inhibitor of MEK that is FDA-approved for the treatment of melanoma. Trametinib effectively blocked the colony formation of immortalized human astrocytes expressing D463H mutant *PRKCA*, while causing only a mild reduction in colony formation for the malignant glioma cell lines with wild-type *PRKCA* alleles (Fig. 4).

**Discussion**

Together, these results identify *PRKCA* as a novel, recurrently mutated oncogene in human cancer. *PRKCA* D463H mutation appears to define chordoid glioma of the third ventricle and genetically distinguishes it from all other brain tumor types that have been studied to date. Thus, evaluation for this mutation may help to distinguish chordoid glioma from other tumor entities during pathologic assessment of suprasellar and intraventricular neoplasms that are difficult to classify based on histologic features alone.

The protein kinase C (PKC) family has been intensely investigated in the context of cancer ever since the discovery that it is a receptor for phorbol esters, the potent tumor promoting compounds derived from the seed oil of the *Croton tiglium* plant[14].

**Fig. 1** Chordoid glioma of the third ventricle is defined by *PRKCA* D463H mutation that localizes in the active site of the kinase domain in the encoded protein kinase C alpha (PKCα). **a** T1-weighted post-contrast magnetic resonance image of a chordoid glioma. **b** Hematoxylin and eosin stained section of a chordoid glioma. Scale bar, 40 μm. **c** Diagram of human PKCα with the location of the D463H mutation identified in each of the 13 chordoid gliomas shown. **d** Amino acid sequence for PKCα in the region of D463 (highlighted in yellow) demonstrating a high degree of conservation from *Homo sapiens* to *C. elegans*. **e** Structural model of the active site of the kinase domain for PKCα showing how the side chain of amino acid Asp463 sits within the ATP binding pocket, where it functions as a proton acceptor during the ATP hydrolysis reaction. Substitution of Asp with His at this codon (D463H) is structurally predicted to result in a side chain with a proton acceptor at nearly the same location, which may potentially alter kinase function through steric differences or ability of the side chains to be protonated or deprotonated based on differential proton affinities

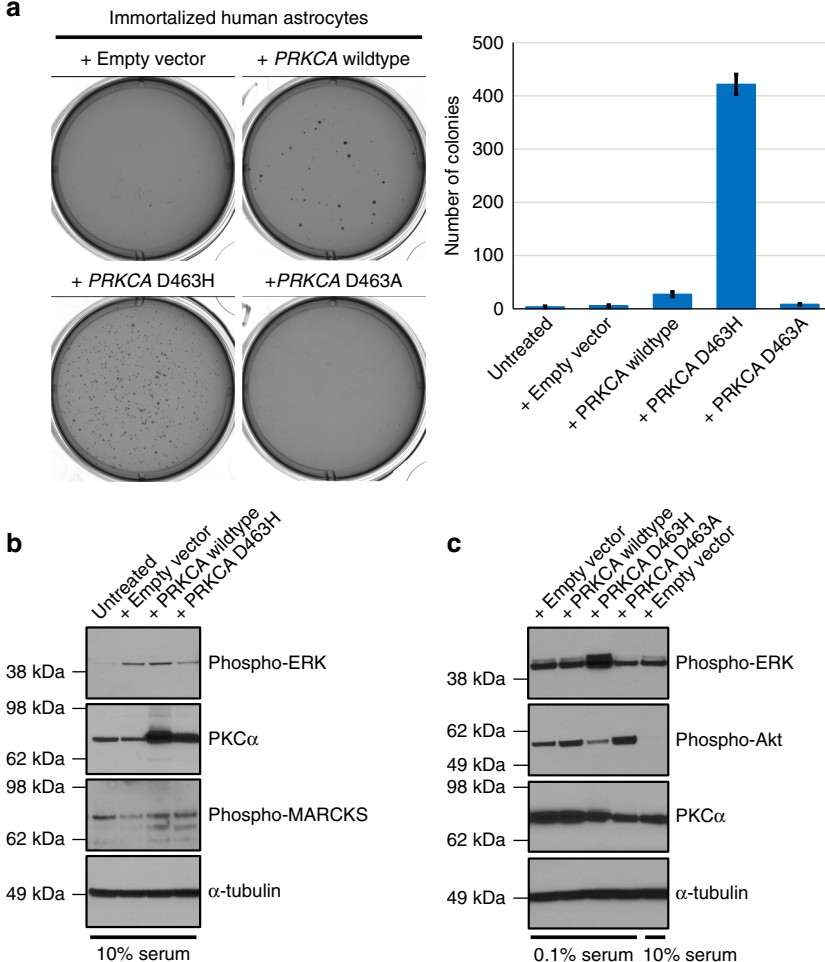

**Fig. 2** Mutant *PRKCA* induces ERK phosphorylation and is sufficient to drive anchorage-independent growth of immortalized human astrocytes. **a** Colony formation in soft agar of immortalized human astrocytes after infection with lentivirus expressing empty vector, wildtype *PRKCA*, D463H tumor mutant *PRKCA*, or D463A non-tumor mutant *PRKCA*. Images of representative wells (left) and quantitation of colony number per well (right) are shown. Error bars represent standard deviation from the mean of six replicates derived from two independent experiments performed in triplicate. **b** Western blots on total cell lysate from 293 T cells growing in 10% fetal bovine serum after transient transfection with empty vector, *PRKCA* wild-type, or *PRKCA* D463H mutant. **c** Western blots on immortalized human astrocytes after lentiviral transduction and then serum starvation for 24 h before collection of total cell lysate

This led to the hypothesis that the PKC genes are proto-oncogenes whose activation by phorbol esters, endogenous ligands, or genetic alteration promotes tumorigenesis. However, recurrent somatic mutations in the genes encoding the various PKC isoforms have not been found in any of the common human cancer types studied to date. Only a small number of tumors with confirmed somatic nonsynonymous mutations of *PRKCA* are present amongst the greater than 30,000 tumors with available sequencing data for *PRKCA* in the version 81 release of the COSMIC database. These *PRKCA* variants are scattered throughout the gene without clustering in the kinase or other functional domain and are predominantly from cancer types with conspicuously high somatic mutational burden such as melanoma and microsatellite unstable gastrointestinal carcinoma (Supplementary Figure 6), suggesting that a majority of these *PRKCA* variants in tumors other than chordoid gliomas are likely to be passenger or bystander mutations. Our study is the first to demonstrate frequent somatic mutations in a PKC gene in a human cancer subtype.

The *PRKCA* mutations in chordoid gliomas are all heterozygous missense mutations that cluster at a single mutational hotspot within the active site of the kinase domain in the encoded PKCα protein. This genetic pattern of heterozygous missense

mutations that cluster at a mutational hotspot within a critical functional domain is strongly suggestive that these are oncogenic, gain-of-function mutations, as opposed to inactivating, loss-of-function events that are typically truncating mutations scattered throughout a gene and accompanied by loss of heterozygosity. Thus, the genetic evidence indicates that *PRKCA* is likely to function as an oncogene, rather than a tumor suppressor gene, in chordoid gliomas. Additionally, our functional data show that this D463H mutation is an oncogenic mutation sufficient to drive anchorage independent growth in immortalized human astrocytes and NIH-3T3 cells. This is in contrast to the wildtype and kinase-dead D463A mutant isoforms that did not promote anchorage independent growth. Together, we believe this combination of genetic and functional data prove that the D463H variant is an oncogenic, gain-of-function mutation in *PRKCA* that drives chordoid gliomas.

While the precise mechanism by which this *PRKCA* mutation drives gliomagenesis remains to be elucidated, it does result in increased MAP kinase pathway activation that may render chordoid gliomas sensitive to MEK inhibitors. Overexpression of the D463H mutant isoform was not found to increase phosphorylation of MARCKS, one of the known kinase substrates of PKCα. We thus speculate that the *PRKCA* D463H mutation does

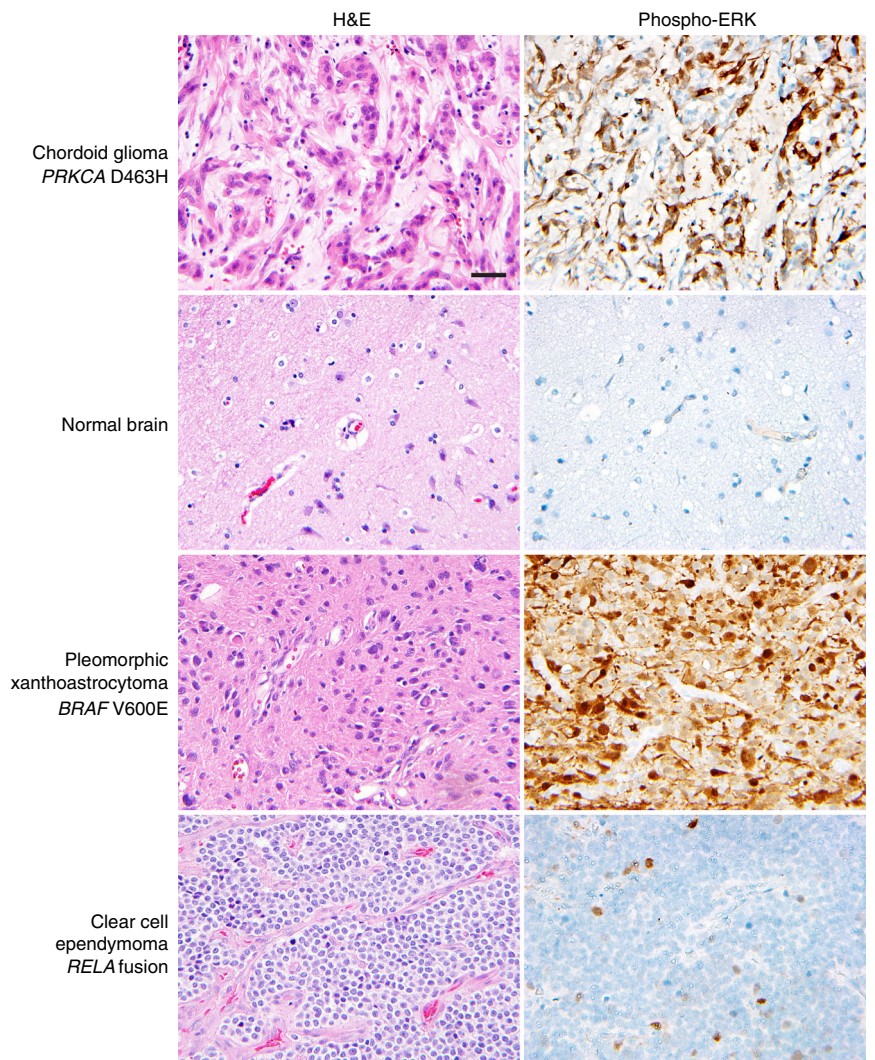

**Fig. 3** Chordoid gliomas demonstrate high levels of phospho-ERK, equivalent to levels seen in other glioma types harboring oncogenic alleles of Ras-Raf-MAP kinase pathway components. Shown are representative images of hematoxylin and eosin (H&E) staining and phospho-ERK immunohistochemistry on a chordoid glioma with *PRKCA* D463H mutation (CG-UVA-1), normal brain, pleomorphic xanthoastrocytoma with *BRAF* V600E mutation, and clear cell ependymoma with *C11orf95-RELA* gene fusion. Scale bar, 40 μm

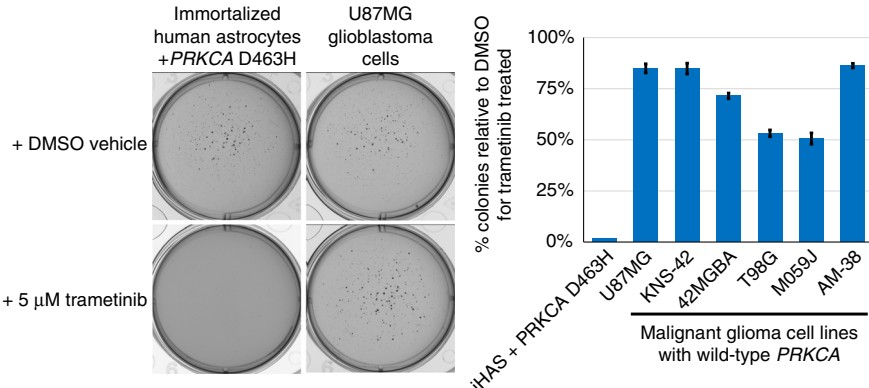

**Fig. 4** MEK inhibition is sufficient to block the oncogenic effects of mutant *PRKCA*. Colony formation in soft agar was assessed for immortalized human astrocytes transduced with D463H mutant *PRKCA* alongside multiple malignant glioma cell lines, all of which harbor wildtype *PRKCA* alleles. DMSO vehicle or 5 μM trametinib was added to the top agar layer during casting. Images of representative wells (left) and quantitation of the percent reduction in colony number of trametinib treated wells relative to DMSO-treated wells (right) are shown. Error bars represent standard deviation from the mean of six replicates derived from two independent experiments performed in triplicate

not function to simply increase the kinase activity of PKCα for its canonical substrates, but rather causes a novel gain of function. Possibilities for this neomorphic function that we speculate include changing the specificity of the kinase activity for new phosphorylation substrates, altering the pH at which the kinase domain is active, or abolishing the kinase activity entirely but promoting tumorigenesis through substrate trapping, acting as a protein scaffold, or other novel kinase-independent mechanism. Additional studies are ongoing in our laboratory to assess the cellular mechanisms by which this mutation fuels tumorigenesis.

In summary, we have elucidated the genetic basis of chordoid glioma of the third ventricle, provided mechanistic insight into how this novel genetic alteration causes this tumor, and identified a potential new therapeutic strategy based on the presence of this defining genetic mutation.

## Methods

**Study population and tumor specimens**. This study was approved by the Committee on Human Research of the University of California, San Francisco, with a waiver of patient consent. Thirteen chordoid gliomas were retrieved from the pathology archives of our respective institutions, spanning years 2001 to 2015. All tumor specimens had been fixed in 10% neutral buffered formalin and embedded in paraffin. Clinical information regarding patient outcomes was obtained from the electronic medical records of our respective institutions. Pathologic re-review of all tumor samples was performed to confirm the diagnosis by two expert neuropathologists (AP and DAS).

**Immunohistochemistry**. Immunohistochemistry was performed on whole formalin-fixed, paraffin-embedded tissue sections using the following antibodies: GFAP (Dako, cat# Z0334, polyclonal, 1:3000 dilution), TTF-1 (Dako, cat# IS056, clone 8G7G3/1, 1:500 dilution), and phospho-p44/42 ERK Thr202/Tyr204 (Cell Signaling, cat #4370, clone D13.14.4E, 1:10,000 dilution). All immunostaining was performed in a Leica Bond-Max automated stainer. Following antigen retrieval, primary antibody was applied for 15 min followed by Bond-Max polymer for 15 min. Diaminobenzidine was used as the chromogen, followed by hematoxylin counterstain.

**Targeted next-generation DNA sequencing and mutational analysis**. Genomic DNA was extracted from formalin-fixed, paraffin-embedded blocks of tumor tissue from 13 patients with chordoid glioma of the third ventricle using the QIAamp DNA FFPE Tissue Kit (Qiagen). Genomic DNA was also extracted from leukocytes in a peripheral blood sample from one of the patients (CG-UCSF-1) and a non-neoplastic gastric biopsy specimen from one of the patients (CG-UCLA-1). Capture-based next-generation DNA sequencing was performed at the University of California, San Francisco Clinical Cancer Genomics Laboratory, using an assay that targets all coding exons of 479 cancer-related genes, select introns of 47 genes, and TERT promoter with a total sequencing footprint of 2.8 Mb (UCSF500 Cancer Panel; Supplementary Data 1)[8]. Sequencing libraries were prepared from genomic DNA, and target enrichment was performed by hybrid capture using a custom oligonucleotide library (Nimblegen SeqCap EZ Choice). Captured libraries were sequenced as paired-end 100 bp reads on an Illumina HiSeq 2500 instrument. Duplicate sequencing reads were removed computationally to allow for accurate allele frequency determination and copy number calling. The analysis was based on the human reference sequence (NCBI build 37) using the following software packages: BWA: 0.7.13, Samtools: 1.1 (using htslib 1.1), Picard tools: 1.97 (1504), GATK: Appistry v2015.1.1–3.4.46–0-ga8e1d99, CNVkit: 0.7.2, Pindel: 0.2.5b8, SATK: Appistry v2015.1.1–1-gea45d62, Annovar: v2016Feb01, Freebayes: 0.9.20, and Delly: 0.7.2[15–22]. Single nucleotide variants and insertions/deletions were visualized and verified using Integrated Genome Viewer. Genome-wide copy number analysis based on on-target and off-target reads was performed by CNVkit and Nexus Copy Number (Biodiscovery)[18].

**PKCα structural modeling**. The crystal structure of the kinase domain for human PKCα has been previously resolved (pdb: 4RA4)[23], except for a loop corresponding to amino acids 617–633. A structural model incorporating this missing loop was built using Protein Local Optimization Program (PLOP)[24] and homology with the human protein kinase A (PKA) crystal structure bound to a non-hydrolysable ATP analog (pdb: 3O7L)[25], which was subjected to a short minimization (steepest descent). A model ATP was then built into the minimized PKCα by modifying the AMP-PNP molecule. D463 from the resulting model was mutated in the pdb file to D463H protonated at the epsilon position. Figures were produced using UCSF Chimera, developed by the Resource for Biocomputing, Visualization, and Informatics at the University of California, San Francisco (supported by NIGMS P41-GM103311)[26].

**PRKCA cDNA expression vector construction and site-directed mutagenesis**. The pcDNA3-PRKCA wildtype expression vector was obtained from Addgene (plasmid #21232). A human wildtype PRKCA cDNA (CCDS11664) with flanking 5′ BglII and 3′ NotI restriction sites was synthesized by GenScript and cloned into the pCDF1-MCS2-EF1-Puro lentiviral expression vector (System Biosciences). D463H and D463A mutations were engineered into the pcDNA3-PRKCA and pCDF1-PRKCA constructs by site-directed mutagenesis using the QuikChange II XL kit (Stratagene) as directed by the manufacturer. The entire coding sequence of all expression vectors was verified by Sanger sequencing. Primer sequences used for the mutagenesis reactions were as follows:

PRKCA D463H Fwd: 5′-GGAATCATTTATAGGCATCTGAAGTTAGATAAC–3′
PRKCA D463H Rev: 5′-GTTATCTAACTTCAGATGCCTATAAATGATTCC-3′
PRKCA D463A Fwd: 5′-GGAATCATTTATAGGGCTCTGAAGTTAGA TAAC-3′
PRKCA D463A Rev: 5′-GTTATCTAACTTCAGAGCCCTATAAATGATTCC-3′.

**Immortalized human astrocytes and cell culture**. Human astrocytes immortalized by retroviral transduction of hTERT, E6, and E7 proteins were generously provided by Dr. Russ Pieper (University of California, San Francisco)[13]. 293 T and NIH-3T3 cells were obtained directly from ATCC and were maintained in Dulbecco's Modified Eagle Medium (DMEM) supplemented with 10% fetal bovine serum at 37 °C in 5% $CO_2$. Glioblastoma and other malignant glioma cell lines harboring wildtype PRKCA alleles were obtained directly from ATCC (U87MG, T98G, M059J), DSMZ (42MGBA), and Japan Health Sciences Foundation Health Science Research Resources Bank (KNS-42, AM-38) and maintained in DMEM supplemented with 10% fetal bovine serum at 37 °C in 5% $CO_2$.

**Lentiviral production and infection**. Empty pCDF1 vector or pCDF1-PRKCA wildtype and mutant expression vectors were co-transfected into 293 T cells with pVSV-G and pFIV-34N helper plasmids (System Biosciences) using Fugene 6 (Roche) as described by the manufacturer. Virus-containing conditioned medium was harvested 48 h after transfection, filtered, and used to infect recipient cells in the presence of 10 µg/mL polybrene.

**Western blot**. Primary antibodies used were PKCα (Santa Cruz Biotechnology, cat# sc-8393, clone H-7, 1:200 dilution), phospho-p44/42 ERK Thr202/Tyr204 (Cell Signaling, cat #4370, clone D13.14.4E, 1:200 dilution), phospho-MARKCS Ser167/170 (Cell Signaling, cat #8722, clone D13E4, 1:100 dilution), phospho-Akt Ser473 (Cell Signaling, cat #4060, clone D9E, 1:100 dilution), and α-tubulin (Thermo Scientific, cat# 62204, clone DM1A, 1:1000 dilution). Protein was extracted from 293 T cells or immortalized human astrocytes in RIPA buffer, resolved by SDS–PAGE, and immunoblotted following standard biochemical techniques. Complete uncropped blots are shown in Supplementary Figure 10.

**Soft agar colony forming assay**. PRKCA wildtype and mutant expressing lentivirus was added to exponentially growing NIH-3T3 cells or immortalized human astrocytes in the presence of 10 µg/mL polybrene. At 24 h following infection, 20,000 cells were suspended in 0.3% agar in DMEM + 10% fetal bovine serum and cast onto a bottom layer of 0.6% agar in DMEM+10% fetal bovine serum in six-well tissue culture plates. For a subset of the experiments, DMSO vehicle or trametinib (Selleck Chemicals) was added to the top agar layer to a final concentration of 5 µM during casting. Soft agar plates were incubated in a humidified chamber at 37 °C containing 5% $CO_2$ for 14 days. Colonies were imaged and counted on a GelCount Colony Counter (Oxford Optronix).

**Reverse transcription–PCR analysis of PRKCA expression**. Total RNA was isolated from 293 T cells at 48 h after transfection with wildtype or D463H mutant PRCKA cDNA constructs using TRIzol (Life Technologies) according to the manufacturer's instructions. Reverse transcription–PCR (RT–PCR) was performed on the total RNA using SuperScript III One-Step RT–PCR System (Life Technologies) to amplify PRKCA mRNA transcripts. RRT–PCR products were sequenced using BigDye terminator chemistry (Applied Biosystems) following standard techniques. Primer sequences used for RT–PCR amplification and sequencing of PRKCA mRNA transcripts were as follows:

PRKCA RT–PCR Fwd: 5′-CCTCATGTACCACATTCAGCA-3′
PRKCA RT–PCR Rev: 5′-TCTGGGGCGATATAATCTGG-3′.

**Data availability**. Sequence data from the 13 chordoid gliomas have been deposited at the European Genome-Phenome Archive (EGA) under accession number EGAS00001002733. All PRKCA expression constructs used in this study have been deposited with Addgene for public distribution. All other remaining data are available within the Article and Supplementary Files, or available from the authors upon request.

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

## Acknowledgements

This study was supported by NIH Director's Early Independence Award (DP5 OD021403) and Career Development Award from the UCSF Brain Tumor SPORE (P50 CA097257) to D.A.S. We thank Dr. Russ Pieper for generously contributing the immortalized human astrocytes used in this study.

## Author contributions

B.G. and M.H. performed the DNA extractions, *PRKCA* expression vector construction, site-directed mutagenesis, and in vitro functional studies. G.M. performed the soft agar colony forming assays and RT–PCR analysis. D.G.R. and M.P.J. performed structural modeling analysis. Y.-H.L. and A.P.W. performed proteomics analysis. N.M.J., J.V.Z., C.O., E.T., J.P.G., I.Y., B.C.B., and D.A.S. performed the targeted next-generation sequencing and genomic analysis. I.H.H., M.S., D.J.B., B.K.K.-D.M., F.J.R., D.N.L., W.H. Y., M.B.L., M.K.R., N.B., T.T., A.W.B., A.P., and D.A.S. procured tumor specimens and performed pathologic assessment. J.J.P. performed immunohistochemistry on tumor tissue. D.A.S. conceptualized the study, reviewed all data, and prepared the figures. All authors assisted with drafting and critically revising the manuscript.

## Additional information

**Competing interests:** The authors declare no competing financial interests.

