## [Peer Review File · Nature Communications]

Reviewers' comments:

Reviewer #1 (Remarks to the Author):

The manuscript by Goode et al., reports the discovery of a novel mutation associated with Choroid Glioma of the third ventricle.

The manuscript is well written and presented data are convincing and novel.

Major comments:

1. The authors suggest that the observed increased proliferation signaling may be done through the MAPK pathway. If so, I suggest showing that MAPK and Phospho-AKT are:
 - a. Increased by performing IHC of FFPE of tumor specimen
 - b. Show that increased phosphor AKT and MAPK decreased in adjacent normal.
 - c. This increase in Phospho AKT and MAPK is specific (or higher expressed) in Choroid tumors and not in another subset of brain tumors
 - d. Increased Phospho AKT and MAPK decreases following treatment with MEK inhibitors.

Minor comment:

1. Page 3, line 46: I am not sure this paper shows that PRKCA is a novel mutation in "human cancer" ... but rather in a subset of rare CNS cancers

Reviewer #2 (Remarks to the Author):

This manuscript by Goode et al establishes a quasi-diagnostic role of PRKCA hotspot mutations in chordoid gliomas.

This finding is novel and relevant and as such worthwhile to be reported. However, before publication of this manuscript should be considered, a few issues should be addressed:

- 1.) "identify a promising new therapy" in the abstract is clearly an overinterpretation of the data provided. Based on the data, at Maximum the authors should refer to this as "potential therapeutic vulnerability".
- 2.) It is surprising that a micro-molar concentration of Trametinib is necessary to see this effect. Did the authors also test at lower concentrations? Did they otherwise check for off-target effects?

Reviewer #3 (Remarks to the Author):

The manuscript by Goode et al describes an exciting finding where a recurring PRKCA mutation (D463H) is found to be causal in Chordoid Glioma. The mutant is heterozygous with an allele frequency ranging from 13-42%. Expressing the D463H mutant promotes anchorage independent growth in immortalized human astrocytes and NIH3T3 cells consistent with this being an oncogenic mutation that confers tumorigenic phenotypes. The authors then demonstrate that expression of the D463H mutant promotes activation of the MAPK pathway and suppresses signaling in the PI3K/AKT pathway and that astrocytes expressing the mutant are sensitive to treatment with a MEK inhibitor. This is an important finding that needs to be conveyed to a broad audience, however several deficiencies within the manuscript need to be addressed prior to publication as follows:

1. The authors refer to the D463H as an “activating kinase domain mutation” without ever showing that the mutation confers an increase in kinase activity. This is a serious deficiency that needs to be addressed. The authors need to perform the following:

a. An in vitro kinase assay with either IPed or purified PKCa mutants that include WT PKCa, D463H, and D463A

b. The authors need to evaluate phosphorylation of direct PKCa substrates in cells overexpressing these mutants, such as MARCKS. What is the phosphorylation status of MARCKS or pan PKC substrate antibody in cells expressing WT PKCa, D463H, and D463A?

2. A mutation at this exact residue was previously characterized (D463N – Antal CE et al, Chem biology, 2014), and defined to be in a constitutively open confirmation, have decreased phosphorylation, and increased degradation. It is essential the authors examine by western blot the PKCa levels in patient samples. It is highly plausible that no or very little PKCa is expressed at the protein levels indicating this would be a loss-of-function mutation. The phosphorylation status and protein levels of the endogenous mutant from the tumor cells from the patients also needs to be examined and included in this manuscript. It appears they already have evidence the mutant D463H expresses at lower levels compared to WT PKCa, Figure 2b.

a. It is difficult to interpret figure 2C as it does not appear that PKCa is expressed above the level of endogenous PKCa in these cells? This is essential if they want to claim this mutant is activating towards the MEK/ERK pathway.

b. A time course experiment needs to be conducted examining the stability of the WT PKCa, the D463H, and the D463A as expressed in cells overtime at 24, 48, 72, and 96 hour time points.

3. Cells from chordoid patients harboring the mutant allele should be depleted of the D463N to show decreased proliferation to confirm this is an essential driver mutation. Correcting the D463N mutant back to WT (via CRISPR/CAS) in chordoid cells from the patients to show a loss of tumorigenic phenotypes would be the most convincing and compelling approach.

Minor comments

1. A western blot should be included for supplemental Figure 8 displaying expression levels of PKCa and downstream substrates MARCKS in NIH3T3 cells.

2. Figure 2C should indicate ERK rather than “MAPK” as this could refer to JNK or P38. What is the phosphorylation status of MEK in these cells?

3. The authors should discuss their findings in light of the recent findings that in general mutations in PKC in cancer are LOF.

A specific point-by-point response to each of the reviewers' comments is below:

Reviewer #1

1. *"The authors suggest that the observed increased proliferation signaling may be done through the MAPK pathway. If so, I suggest showing that MAPK and Phospho-AKT are:*
 - a. *Increased by performing IHC of FFPE of tumor specimen.*
 - b. *Show that increased phosphor AKT and MAPK decreased in adjacent normal.*
 - c. *This increase in Phospho AKT and MAPK is specific (or higher expressed) in Chordoid tumors and not in another subset of brain tumors.*
 - d. *Increased Phospho AKT and MAPK decreases following treatment with MEK inhibitors."*

As suggested by this reviewer, we have studied the levels of phospho-ERK in chordoid glioma tumor samples by immunohistochemistry. We found high levels of phospho-ERK in chordoid gliomas, equivalent to the levels seen in other gliomas harboring known genetic alterations within the Ras-Raf-MAP kinase signaling pathway (e.g. pilocytic astrocytoma with *KIAA1549-BRAF* gene fusion and pleomorphic xanthoastrocytoma with *BRAF* p.V600E mutation). This phospho-ERK immunostaining was specific to the tumor cells in chordoid gliomas, as the non-neoplastic endothelial cells and lymphocytes were negative. Furthermore, we demonstrate that non-neoplastic brain parenchyma and gliomas driven by genetic alterations not in the MAP kinase pathway (e.g. clear cell ependymoma driven by *RELA* gene fusion and NF- κ B pathway activation) lack significant phospho-ERK positivity relative to that seen in chordoid gliomas. We believe that this provides additional functional evidence that the *PRKCA* p.D463H mutation drives chordoid glioma, at least in part, by activation of the MAP kinase signaling pathway. These new studies have been added to the revised manuscript in Figure 3.

2. *"Page 3, line 46: I am not sure this paper shows that PRKCA is a novel mutation in "human cancer" ... but rather in a subset of rare CNS cancers"*

We agree our study only shows that *PRKCA* is recurrently mutated in a rare subtype of CNS cancer. However, we would also like to point out the several decades of research that have investigated a potential role of protein kinase C (PKC) in the pathogenesis of human cancer. Following the discovery in the 1980's that PKC functions as the intracellular receptor for phorbol esters, a potent tumorigenic compound derived from the seed oil of the *Croton tiglium* plant, it has since been postulated that deregulated PKC signaling is important in human cancer. However, recurrent somatic mutations in the genes encoding the various PKC isoforms have not been found in any of the common human cancer types studied to date. Our study is the first to demonstrate recurrent somatic mutations in a PKC gene in human cancer. Thus, our study has broad impact for the large body of research that has studied PKC signaling in human tumorigenesis, with greater than 11,000 reports currently archived in PubMed that include both the terms cancer and protein kinase C.

Reviewer #2

1. *"'Identify a promising new therapy' in the abstract is clearly an overinterpretation of the data provided. Based on the data, at maximum the authors should refer to this as "potential therapeutic vulnerability".*

We agree and have modified the abstract as suggested in the revised manuscript.

2. "It is surprising that a micro-molar concentration of Trametinib is necessary to see this effect. Did the authors also test at lower concentrations? Did they otherwise check for off-target effects?"

The concentration of small molecule inhibitors used in colony forming assays in soft-agar is generally an order of magnitude higher than that used during in vitro culture for adherent cells grown as a monolayer covered with liquid media. This stems from the fact that in soft-agar the inhibitor is suspended and not freely diffusible, nor is it possible to add additional drug over the two to three week time course of the experiment. For many small molecule inhibitors that are effective at nanomolar concentrations for adherent monolayer cell cultures, including vemurafenib and trametinib, these drugs are typically used at 1-10 micromolar concentrations in soft-agar assays [refs. 1-3]. We do not believe that there are significant off-target effects of the trametinib in our soft-agar assays, as at the concentration used to completely block colony formation in immortalized human astrocytes transduced with D463H mutant PKC α , there was very little inhibition of colony formation by malignant glioma cell lines harboring wildtype *PRKCA* alleles.

References:

1. Shen CH, et al. Loss of cohesin complex components STAG2 or STAG3 confers resistance to BRAF inhibition in melanoma. *Nature Medicine* 22: 1056-1061, 2016.
2. Escuin-Ordinas H, et al. COX-2 inhibition prevents the appearance of cutaneous squamous cell carcinomas accelerated by BRAF inhibitors. *Molecular Oncology* 8: 250-260, 2014.
3. Yan C, et al. Discovery and characterization of small molecules that target the GTPase Ral. *Nature* 515: 443-447, 2014.

Reviewer #3

1. "The authors refer to the D463H as an "activating kinase domain mutation" without ever showing that the mutation confers an increase in kinase activity. This is a serious deficiency that needs to be addressed. The authors need to perform the following:

a. An in vitro kinase assay with either IPed or purified PKC α mutants that include WT PKC α , D463H, and D463A

b. The authors need to evaluate phosphorylation of direct PKC α substrates in cells overexpressing these mutants, such as MARCKS. What is the phosphorylation status of MARCKS or pan PKC substrate antibody in cells expressing WT PKC α , D463H, and D463A?"

We agree that our results do not demonstrate that the *PRKCA* D463H mutation we have discovered in chordoid gliomas leads to an increase in kinase activity. We had used the term "activating" in our initial submission to describe that this recurrent mutation located in the active site of the kinase domain of PKC α is an oncogenic gain-of-function variant, based on the fact that we found the D463H mutant isoform promoted anchorage independent growth of multiple human cell lines, while the wildtype and D463A kinase-dead mutants did not. As suggested, we have added Western blot data for phospho-MARCKS to the revised manuscript, which demonstrates that expression of D463H mutant PKC α does not lead to an increase in phospho-MARCKS, one of the known phosphorylation substrates for protein kinase C. We believe that these data suggest that the *PRKCA* D463H mutation is likely to be promoting the development of chordoid gliomas through a novel gain-of-function, perhaps through changing the specificity of the kinase activity for novel phosphorylation substrates. This hypothesis is based on the fact that the mutations recur at this single codon and that unequivocal loss-of-function mutations such as frameshift or nonsense mutations in *PRKCA* were not detected in any tumors, making it unlikely that the mutations are loss of function variants. To avoid any confusion, we have removed changed the title

of the revised manuscript to “A recurrent kinase domain mutation in *PRKCA* defines chordoid glioma of the third ventricle”, replacing the word activating with recurrent.

2. “A mutation at this exact residue was previously characterized (D463N – Antal CE et al, *Chem biology*, 2014), and defined to be in a constitutively open confirmation, have decreased phosphorylation, and increased degradation. It is essential the authors examine by western blot the PKCa levels in patient samples. It is highly plausible that no or very little PKCa is expressed at the protein levels indicating this would be a loss-of-function mutation. The phosphorylation status and protein levels of the endogenous mutant from the tumor cells from the patients also needs to be examined and included in this manuscript. It appears they already have evidence the mutant D463H expresses at lower levels compared to WT PKCa, Figure 2b.

a. It is difficult to interpret figure 2C as it does not appear that PKCa is expressed above the level of endogenous PKCa in these cells? This is essential if they want to claim this mutant is activating towards the MEK/ERK pathway.

b. A time course experiment needs to be conducted examining the stability of the WT PKCa, the D463H, and the D463A as expressed in cells overtime at 24, 48, 72, and 96 hour time points.”

Unfortunately chordoid glioma of the third ventricle is a rare tumor entity, and cell lines from patient samples have not been established to date by any investigators. As the only tumor tissue available for the chordoid gliomas included in this study is formalin-fixed and paraffin-embedded (FFPE), it is not possible to study the levels of PKC α in chordoid gliomas by Western blot as suggested by this reviewer. Alternatively, immunohistochemistry is the principle technique used for studying protein expression in FFPE tissue. However, immunohistochemistry is actually a notoriously unreliable method for studying minor differences in protein expression levels (e.g. two-fold or less) in primary tumor tissue due to technical variability between samples such as differences in post-operative ischemia time prior to formalin fixation, amount of time in formalin fixative prior to embedding, antigen retrieval methods, antibody dilution, etc. Additionally, as there is not currently wildtype-specific and mutant-specific antibodies available for PKC α , immunohistochemistry for total PKC α on the FFPE tumor tissue of our chordoid glioma patient samples would not be able to differentiate between wildtype and mutant proteins, making this an ineffective method for addressing the question raised by this reviewer.

However, the main concern raised by the reviewer is regarding the alternative explanation that this D463H mutation may be a loss-of-function mutation. We believe that we have conclusively documented in our manuscript through a combination of genetic and functional data that this is an activating oncogenic mutation in *PRKCA* as already discussed in detail above.

Regarding the stability of the mutant isoform of PKC α , this is an interesting point raised by the reviewer that is challenging to address. While robust overexpression of wildtype and mutant isoforms of PKCa was seen after transient transfection in 293T cells, lentiviral transduction of wildtype and mutant isoforms of *PRKCA* in immortalized human astrocytes and NIH-3T3 cells did not lead to a detectable increase in PKC α protein levels, possibly due to only low levels of protein overexpression that are beyond the limits of detection by Western blot using chemiluminescence. Alternatively, it may also be possible that stably expressing immortalized human astrocytes and NIH-3T3 cells after lentiviral transduction have auto-regulatory feedback mechanisms controlling the expression of PKC α , thus resulting in absence of the overexpression seen after transient transfection in 293T cells. Regardless of our ability to detect significant protein overexpression, lentiviral transduction of both immortalized human astrocytes and NIH-3T3 cells with D463H mutant, but not wildtype or D463A kinase-dead *PRKCA*, consistently led to colony formation in soft agar (i.e. anchorage independent growth) that was observed

in multiple replicates in multiple independent experiments performed on different days. Thus, it is not possible for us to study PKC α levels after lentiviral transduction in immortalized human astrocytes, and the applicability of studying PKC α levels after transient transfection in 293T cells would not really be simulating a meaningful biologic scenario. Furthermore, the results of this experiment would be very difficult to interpret given the known variability in expression levels due to variable sample to sample transfection efficiency.

3. *“Cells from chordoid patients harboring the mutant allele should be depleted of the D463N to show decreased proliferation to confirm this is an essential driver mutation. Correcting the D463N mutant back to WT (via CRISPR/CAS) in chordoid cells from the patients to show a loss of tumorigenic phenotypes would be the most convincing and compelling approach.”*

Unfortunately chordoid glioma of the third ventricle is a rare tumor entity, and no such cell lines have been established to date by any investigators for this tumor type. While we agree this suggested avenue of experimentation would be very interesting and informative to pursue, this is unfortunately not feasible at this time. Thus, we chose to use overexpression of wildtype and mutant *PRKCA* cDNA in immortalized human astrocytes (as well as NIH-3T3 cells) to study the function of the D463H mutation during tumorigenesis. We believe that the data presented in our revised manuscript provide convincing evidence that the D463H mutation is an oncogenic driver in chordoid gliomas.

4. *“A western blot should be included for supplemental Figure 8 displaying expression levels of PKCa and downstream substrates MARCKS in NIH3T3 cells.”*

Similar to the immortalized human astrocytes, lentiviral transduction of wildtype and mutant isoforms of *PRKCA* in NIH-3T3 cells did not lead to a detectable increase in PKC α protein levels, possibly due to only low levels of protein overexpression that are beyond the limits of detection by Western blot using chemiluminescence. Alternatively, it may also be possible that stably expressing NIH-3T3 cells after lentiviral transduction have auto-regulatory feedback mechanisms controlling the expression of PKC α , thus resulting in absence of the overexpression seen after transient transfection in 293T cells. Regardless of our ability to detect significant protein overexpression, lentiviral transduction of NIH-3T3 cells with D463H mutant but not wildtype *PRKCA* consistently led to colony formation in soft agar (i.e. anchorage independent growth) that was observed in multiple replicates in multiple independent experiments performed on different days. Similarly, no increase in phospho-MARCKS was detected in NIH-3T3 cells. We have chosen not to include these negative data in this supplemental figure since they are redundant with the data present in the main figures.

5. *“Figure 2C should indicate ERK rather than ‘MAPK’ as this could refer to JNK or P38. What is the phosphorylation status of MEK in these cells?”*

We agree and have modified the figure as suggested in the revised manuscript.

6. *“The authors should discuss their findings in light of the recent findings that in general mutations in PKC in cancer are LOF.”*

As suggested, we have added additional text to our revised manuscript regarding the oncogenic function of the D463H mutation identified in our study, in contrast to prior reports that have suggested that the rare somatic mutations in *PRKCA* and other PKC genes found predominantly in tumor types with a very

high somatic mutation burden such as melanoma and microsatellite unstable gastrointestinal carcinomas are inactivating/loss-of-function mutations.

Reviewers' comments:

Reviewer #1 (Remarks to the Author):

The authors have addressed all of my concerns and I have no further concerns.

Reviewer #2 (Remarks to the Author):

My concerns were adequately addressed.

Reviewer #3 (Remarks to the Author):

In the manuscript by Goode et al, unfortunately the authors have done very little to address the critical concerns of the reviewers and I will not be able to recommend that this manuscript be published until the authors address these concerns.

The main issue that has not been addressed is how this mutation will affect the catalytic activity of PKCa. This can be determined through routine biochemical analysis using in vitro and in cell kinase assays that are properly controlled to determine the impact of this mutation on PKC kinase activity as outlined in the initial review. Without this data, the authors must refer to the publication from Peter Parker's lab (Cameron et al NSMB 2009) supplemental figure 2A where they unequivocally demonstrated that mutation at D463 (to A or N) abolishes catalytic activity. I have uploaded this supplemental data for review of the editors and this is complementary to the work of Antal et al that demonstrates that mutation at D463 has decreased stability and phosphorylation as discussed in the initial review. Without this data, the authors would need to reference these previous publications and state that this mutation likely results in complete loss of catalytic activity (a loss-of-catalytic activity) mutation, that has neomorphic properties as possibly a scaffolding protein that may result in kinase-independent activation of other pathways.

Therefore, the authors must remove the following since it is unproven and ambiguous:

Line 130:

"We speculate that this substitution may alter the activity or specificity of the kinase function by perturbing the ATP hydrolysis reaction, potentially in a pH-dependent manner" – there is absolutely zero data to support this speculation that the mutation would alter activity or substrate specificity (for the later a mutant pull-down followed by mass spec to identify a unique interactome would be required).

Line 146:

"These results indicate that p.D463H is an activating, oncogenic mutation." – this is undoubtedly an oncogenic mutation, but there is zero data to support this is an "activating" mutation.

Line 149:

"PRKCA D463H mutation induces activation of the MAP kinase signaling pathway" – the combined data do not support this statement. There is no increases in 293T cells where the mutant is convincingly expressed on ERK phosphorylation at 10% FBS - what is the effect at 0.1% FBS in 293T cells? There is no way to tell the mutants are even expressed in the astrocytes – at the very least qPCR needs to be performed to demonstrate the mutant is expressed and at higher levels than endogenous PKC alpha. The endogenous levels are extremely high in these astrocytes as seen in the empty vector control and it may be difficult to express PKC exogenously much above endogenous levels and the results could be artifacts.

Line 173:

"Thus, PRKCA D463H mutation is likely to drive chordoid glioma, at least in part, by activation of the MAP kinase signaling pathway." – Again this is unproven and there is a chance that the mutant is suppressing AKT pathway activation, which will lead to ERK pathway activation.

Line 208:

"This genetic pattern of heterozygous missense mutations that cluster at a mutational hotspot within a critical functional domain is strongly suggestive that these are activating, gain-of-function mutations, as opposed to inactivating, loss-of-function events that are typically truncating mutations scattered throughout a gene and accompanied by loss of heterozygosity." Shockingly the authors make this claim with zero biochemical data to support this claim, even though it is extremely feasible to test if this is an "activating" gain-of-function mutation as opposed to a neomorphic loss of catalytic activity mutation.

Line 213:

"Additionally, our functional data show that this D463H mutation is an activating oncogenic mutation" – "activating" needs to be removed since no data is provided to indicate this is an "activating" mutation.

Minor comments:

1. PKC alpha overexpression in 293T cells should lead to a robust increase in MARCKS phosphorylation as shown by many labs. This suggests the WT construct is not WT and may harbor a loss of catalytic activity mutation, which would bring into question other results presented in the manuscript.
2. The authors state that there is likely "only low levels of protein overexpression" in the astrocytes, how is it possible that such low levels of "over expression" will cause such a dramatic change in colony formation? This unlikely and again may be artifactual. Why wasn't an inducible (dox or tet) system used?

Reviewer #3

The main issue that has not been addressed is how this mutation will affect the catalytic activity of PKCa. This can be determined through routine biochemical analysis using in vitro and in cell kinase assays that are properly controlled to determine the impact of this mutation on PKC kinase activity as outlined in the initial review. Without this data, the authors must refer to the publication from Peter Parker's lab (Cameron et al NSMB 2009) supplemental figure 2A where they unequivocally demonstrated that mutation at D463 (to A or N) abolishes catalytic activity. I have uploaded this supplemental data for review of the editors and this is complementary to the work of Antal et al that demonstrates that mutation at D463 has decreased stability and phosphorylation as discussed in the initial review. Without this data, the authors would need to reference these previous publications and state that this mutation likely results in complete loss of catalytic activity (a loss-of-catalytic activity) mutation, that has neomorphic properties as possibly a scaffolding protein that may result in kinase-independent activation of other pathways.

We thank the reviewer for his/her thoughtful comments on our manuscript and suggestions for improvement. As requested, we have referenced the Cameron et al NSMB 2009 study in the revised manuscript, as well as discussed the unique possibility that this D463H mutation may result in a loss of catalytic activity but still promote oncogenesis due to a neomorphic gain-of-function that the kinase dead D463A mutant does not harbor, possibly through substrate trapping, acting as a scaffolding protein, or other novel kinase-independent mechanism. We also agree that in vitro kinase assays are a critical next step in determining the mechanism(s) by which this mutation causes chordoid gliomas and are currently working to develop collaborations to assist our laboratory with these types of experiments. However, we feel that this study is sufficiently novel, clinically relevant, and well developed without these in vitro biochemical assays to warrant publication in its present form in *Nature Communications*. Please find our comments below to each of your specific concerns. We sincerely hope that we have been successful in addressing each of these points of contention that you have raised.

Therefore, the authors must remove the following since it is unproven and ambiguous:

Line 130 of Results: "We speculate that this substitution may alter the activity or specificity of the kinase function by perturbing the ATP hydrolysis reaction, potentially in a pH-dependent manner" – there is absolutely zero data to support this speculation that the mutation would alter activity or substrate specificity (for the later a mutant pull-down followed by mass spec to identify a unique interactome would be required).

We agree that this statement was purely speculation and, as such, is inappropriate for the Results section of our manuscript. We have therefore deleted this sentence from the manuscript as requested and replaced with a statement that we feel is more appropriate for the Results section – "This substitution is therefore likely to alter the activity of the kinase function by perturbing the ATP hydrolysis reaction."

Line 146 of Results: "These results indicate that p.D463H is an activating, oncogenic mutation." – this is undoubtedly an oncogenic mutation, but there is zero data to support this is an "activating" mutation.

We agree that the data in our manuscript only show that this is an oncogenic, gain-of-function mutation but do not demonstrate that it activates or enhances the kinase activity of the PKC α protein. As such, we have deleted the word “activating” from this sentence as requested.

Line 149 of Results: “PRKCA D463H mutation induces activation of the MAP kinase signaling pathway” – the combined data do not support this statement. There is no increases in 293T cells where the mutant is convincingly expressed on ERK phosphorylation at 10% FBS - what is the effect at 0.1% FBS in 293T cells? There is no way to tell the mutants are even expressed in the astrocytes – at the very least qPCR needs to be performed to demonstrate the mutant is expressed and at higher levels than endogenous PKC alpha. The endogenous levels are extremely high in these astrocytes as seen in the empty vector control and it may be difficult to express PKC exogenously much above endogenous levels and the results could be artifacts.

We agree that our data do not show a direct causation between *PRKCA* mutation and activation of the MAP kinase signaling pathway. We consider it not only possible, but perhaps even likely, that this may be an indirect effect of the mutation through an as yet unidentified mechanism. We have made changes in both the Results and Discussion sections of the revised manuscript to more clearly account for this possibility. We have also added a new Supplemental Figure 9 to the revised manuscript wherein we performed RT-PCR to provide confirmation that our ectopic expression of wildtype and D463H mutant cDNA is leading to expression of wildtype and D463H mutant *PRKCA* mRNA transcripts.

Line 173 of Results: “Thus, PRKCA D463H mutation is likely to drive chordoid glioma, at least in part, by activation of the MAP kinase signaling pathway.” – Again this is unproven and there is a chance that the mutant is suppressing AKT pathway activation, which will lead to ERK pathway activation.

We have revised this sentence by adding “although this may be a downstream consequence rather than direct phosphorylation by mutant PKC α ” to account for the possibility that activation of the MAP kinase signaling pathway may likely be a downstream consequence of the *PRKCA* mutation rather than a direct effect.

Line 208 of Discussion: “This genetic pattern of heterozygous missense mutations that cluster at a mutational hotspot within a critical functional domain is strongly suggestive that these are activating, gain-of-function mutations, as opposed to inactivating, loss-of-function events that are typically truncating mutations scattered throughout a gene and accompanied by loss of heterozygosity.” Shockingly the authors make this claim with zero biochemical data to support this claim, even though it is extremely feasible to test if this is an “activating” gain-of-function mutation as opposed to a neomorphic loss of catalytic activity mutation.

We have removed the word “activating” from this sentence in the Discussion as requested.

Line 213 of Discussion: “Additionally, our functional data show that this D463H mutation is an activating oncogenic mutation” – “activating” needs to be removed since no data is provided to indicate this is an “activating” mutation.

We have removed the word “activating” from this sentence in the Discussion as requested.

Minor comments:

1. PKC alpha overexpression in 293T cells should lead to a robust increase in MARCKS phosphorylation as shown by many labs. This suggests the WT construct is not WT and may harbor a loss of catalytic activity mutation, which would bring into question other results presented in the manuscript.

We also initially shared this concern with the reviewer. However, as stated in the Methods section of the manuscript, the entire coding sequence of all *PRKCA* expression vectors used in our study was verified by Sanger sequencing. Additionally, we have added a new Supplemental Figure 9 to the revised manuscript wherein we provide confirmation that our ectopic expression of wildtype and D463H mutant cDNA is leading to expression of wildtype and D463H mutant *PRKCA* mRNA transcripts. We hope that this alleviates any concerns about the integrity of the *PRKCA* expression vectors used in our study. However, we do not have a good explanation for why our ectopic overexpression of wildtype PKC α protein in 293T cells did not lead to a more robust increase in phospho-MARCKS levels as has been reported in the literature.

2. The authors state that there is likely “only low levels of protein overexpression” in the astrocytes, how is it possible that such low levels of “over expression” will cause such a dramatic change in colony formation? This unlikely and again may be artifactual. Why wasn’t an inducible (dox or tet) system used?

As indicated in our prior submission, we think that a significant increase in PKC α levels was not observed after stable lentiviral transduction in immortalized human astrocytes due to only low levels of overexpression or possibly auto-regulatory feedback mechanisms controlling the expression of PKC α . Our data indicate that D463H mutant *PRKCA* is a potent oncogene, as it causes anchorage-independent growth of both immortalized human astrocytes and NIH-3T3 cells in multiple replicates of multiple independent experiments. We are highly confident in these scientific results, and do not feel that use of inducible system would be substantially more informative.

REVIEWERS' COMMENTS:

Reviewer #3 (Remarks to the Author):

I feel the biochemical data is critical to defining the function of mutant PKCa and the mechanism by which this mutant form of the kinase is promoting tumorigenesis of chordoid glioma of the third ventricle. This is important to understanding the mechanisms of tumorigenesis, not only for this type of glioma, but also for the role of PKCs in proliferative malignancies in general. Overall, functional characterization of genetic mutations in PKCs in proliferative disorders overwhelming point to PKCs being tumor suppressors in cancer (Antal et al, Cell 2015) or inhibiting proliferation and survival of B cells as is the case with causal homozygous PKCd Loss-of-function mutations in lymphoproliferative disorders and JSLE (Kuehn et al, Blood 2013, Salzer et. at. Blood 2013, Belot et al, Arthritis and Rheumatism 2013). This study would provide more critical support for the role of PKCs as suppressors of proliferative disorders and cancer and I would generally be quite surprised if this mutation did not abolish the catalytic activity of PKCa.

However, I feel the authors have changed the text appropriately to somewhat address this concern and I tend to agree with them that the finding is novel enough and clinically relevant to warrant publication.

Reviewer #3

I feel the biochemical data is critical to defining the function of mutant PKCa and the mechanism by which this mutant form of the kinase is promoting tumorigenesis of chordoid glioma of the third ventricle. This is important to understanding the mechanisms of tumorigenesis, not only for this type of glioma, but also for the role of PKCs in proliferative malignancies in general. Overall, functional characterization of genetic mutations in PKCs in proliferative disorders overwhelming point to PKCs being tumor suppressors in cancer (Antal et al, Cell 2015) or inhibiting proliferation and survival of B cells as is the case with causal homozygous PKCd Loss-of-function mutations in lymphoproliferative disorders and JSLE (Kuehn et al, Blood 2013, Salzer et. at. Blood 2013, Belot et al, Arthritis and Rheumatism 2013). This study would provide more critical support for the role of PKCs as suppressors of proliferative disorders and cancer and I would generally be quite surprised if this mutation did not abolish the catalytic activity of PKCa.

However, I feel the authors have changed the text appropriately to somewhat address this concern and I tend to agree with them that the finding is novel enough and clinically relevant to warrant publication.

We thank the reviewer for his/her thoughtful comments on our manuscript and suggestions for improvement throughout this review process. While it is possible that genetic mutations of PKC genes in other cellular and tumor contexts are loss-of-function events, we believe that the data we present in our manuscript prove that the *PRKCA* D463H mutation is an oncogenic, gain-of-function event in chordoid gliomas of the third ventricle. We agree that biochemical characterization (e.g. in vitro kinase assay) is an important next step in determining the precise mechanism by which this D463H mutation in *PRKCA* drives chordoid gliomas.